mechanical engineering

rock mechanics, rock cutting, cutting mechanics, design optimization

**Author for correspondence:**
Zhi-jun Wan
e-mail: zhjwan@cumt.edu.cn

# Investigation of the influence mechanism of rock damage on rock fragmentation and cutting performance by the discrete element method

Si-fei Liu[1,2], Shuai-feng Lu[1,2], Zhi-jun Wan[1,2] and Jing-yi Cheng[1,2]

[1]Key Laboratory of Deep Coal Resource Mining (CUMT), Ministry of Education of China, and [2]School of Mines, China University of Mining and Technology, Xuzhou 221116, People's Republic of China

S-fL, 0000-0001-8331-5355; Z-jW, 0000-0003-2095-6336

Rock damage is one of the key factors in the design and model choice of mining machinery. In this paper, the influence of rock damage on rock fragmentation and cutting performance was studied using $PFC^{2D}$. In $PFC^{2D}$ software, it is feasible to get rock models with different damage factors by reducing the effective modulus, tensile and shear strength of bond by using the proportional factors. A linear relationship was obtained between the proportion factor and damage factor. Furthermore, numerical simulations of rock cutting with different damage factors were carried out. The results show that with the increase of damage factor, the rock cutting failure mode changes from tensile failure to brittle failure, accompanied by the propagation of macro cracks, the formation of large debris and a notable decrease in the peak cutting force. The mean cutting force is negatively correlated with the damage factor. Besides this, the instability of cutting force was evaluated by the fluctuation index and the pulse number of unit displacement. It was found that the cutting force was quite stable when the damage factor was 0.3, which improves the reliability of cutting machines. Finally, the cutting energy consumption of rock cutting with different damage factors was analysed. The results reveal that an increase of damage factor can raise the rock cutting efficiency. The aforementioned findings play a significant role in the development of assisted rock-breaking technologies and the design of cutting head layout of mining machinery.

# 1. Introduction

Over the years, there has been great progress in rock-cutting processes, improving the mining efficiency and accelerating the exploitation of deep layer ore resources. Aiming at optimizing the design of mining machinery, many scholars have focused on the influence of rock properties on rock fragmentation and cutting [1–7]. They revealed that uniaxial compressive strength (UCS), tensile strength, elastic modulus, brittleness index, etc., were important factors for the cutting force. However, the effect of the degree of rock damage on rock cutting was rarely investigated.

Two major working procedures, namely, the excavation of roadways and the mining of ore body, are involved in the mining process, as shown in figure 1. There are two kinds of advanced abutment pressure in front of the working face in the vertical direction [8–10], as shown by curves $T_1$ and $T_2$ in figure 1. $T_1$ indicates that the abutment pressure is lower than the limit load of the ore body, and the ore body is still in the elastic stage without being damaged. By contrast, $T_2$ suggests that the ore body has been partially damaged, and the yielding ore body can still bear the supporting pressure and remain stable by relying on the residual strength. In fact, most ore bodies are affected by the $T_2$ curve, but the existing cutting theory does not take the ore body damage into account. Meanwhile, with the development of mechanical technology, a large number of assisted rock-breaking technologies, such as high-pressure water jet, microwave, electromagnetic radiation and laser technology, have emerged in recent years [11–17], which have improved the rock-breaking efficiency of mining machineries. The working mechanism is to destroy the ore body with the use of assisted rock-breaking technology before the breaking of rock by mechanical breaking equipment. In this way, the ore body is damaged and then mined by mechanical equipment, as presented in figure 2 [18]. Many scholars have researched the significant effects of assisted rock-breaking methods on improving rock-breaking efficiency and reducing cutting force, but they have not mentioned the influence of the degree of ore body damage on cutting performance. Therefore, it is important to study the effect of rock damage on rock cutting. The study can not only optimize the design of excavation machinery, but also provide a quantitative reference for assisted rock-breaking technology.

In order to study the influence of rock damage on the cutting mechanism and effect, in this study the mechanical parameters of rock with different degrees of damage were determined using the discrete element software PFC$^{2D}$. Then, a single-pick cutting model of different damaged rocks was established. Through this model, the rock crushing process and cutting force during the cutting were obtained, and the influence of rock damage on the cutting mechanism was explored. Furthermore, the effect of rock damage on cutting force and cutting energy consumption was studied. In this study, the effects of rock damage on rock fragmentation and rock interception were quantitatively analysed for the first time by numerical simulation.

# 2. Model of rock samples with different degrees of damage

In this study, a rock model was established by using the linear parallel bond model [19]. Parallel bond, distributed evenly on the contact interface, is considered as a series of springs with fixed normal and shear stiffness between contact elements. As exhibited in figure 3a, the rock damage is mainly determined by seven parameters which can be obtained through a uniaxial compression test [20,21]. They are the elastic modulus of linear contact, the stiffness ratio of linear contact, the elastic modulus of parallel bond, the stiffness ratio of parallel bond, the tensile strength of parallel bond, the shear strength of parallel bond and the friction angle.

Rock damage manifests in a decrease in bearing capacity and the appearance of microscopic cracks, and corresponds to microscopic changes in the elastic modulus, the tensile strength and the shear strength. In order to simulate rocks with different damage coefficients, the coal from Zhaolou Coal Mine, China, was selected as the standard coal sample. First, its microscopic parameters such as the elastic modulus, the tensile strength and the shear strength were obtained through a uniaxial compression test, and then a new variable $k$ ($k < 1$) was introduced. $k$ is a proportional factor used to adjust the micro-parameters. When the rock was destroyed, the model classified microcracks into tensile cracks and shear cracks according to whether the maximum stress exceeded the tensile strength or the shear strength of the bond fracture, as shown in figure 3b.

## 2.1. Uniaxial compression tests

In the uniaxial compression test, a standard 50 mm × 100 mm model which consisted of 3600 particles was established as the numerical model. The test was carried out in a static loading mode with a

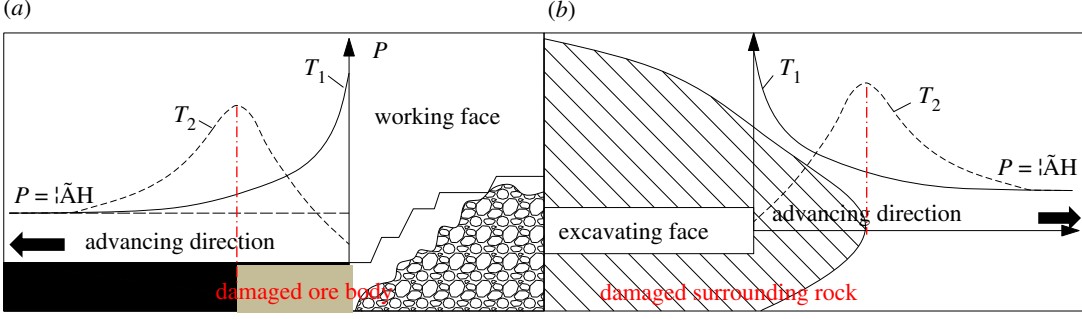

**Figure 1.** Schematic diagram of mining. (*a*) Ore body mining. (*b*) Tunnel excavation.

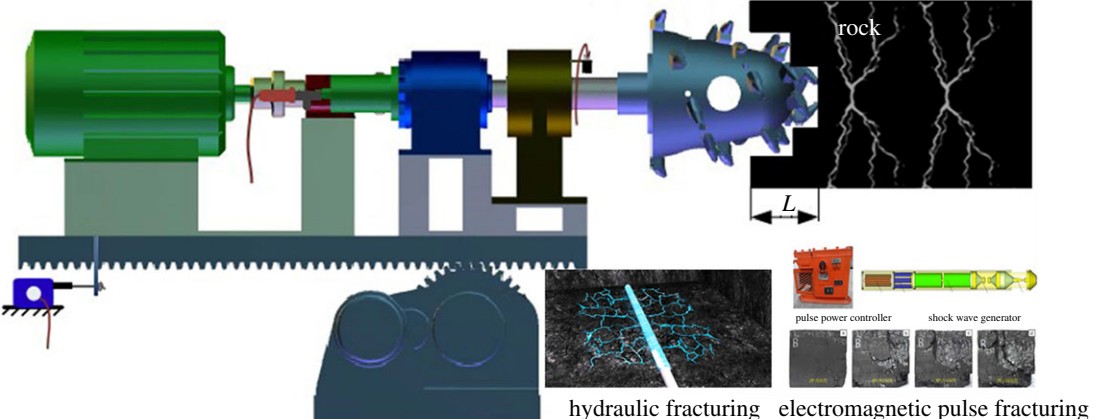

**Figure 2.** Assisted rock-breaking technology [18].

loading speed of 0.005 m s$^{-1}$. For the identity of the model, only the microscopic parameters were changed during the test. The specific microscopic parameters of the standard coal sample in Zhaolou Coal Mine are shown in table 1. The results of fitting between numerical simulation and physical experiments are illustrated in figure 4.

In this paper, rocks with different degrees of damage were obtained by reducing the elastic modulus, tensile strength and shear strength simultaneously according to the proportional factor (*k*). The microparameters of the sample with different values of *k* are listed in table 2. Figure 5 shows the stress–strain curves under different values of *k*. The elastic modulus and UCS changed significantly. That is, the effective supporting capacity dropped under the same strain, which agrees with the concept of rock damage in fracture mechanics. Based on the above theory, it is reasonable to obtain rocks with different degrees of damage through the changes in the proportionality factors.

## 2.2. Rock damage

There are many models [22–25] for calculating rock damage factors. According to Lemaitre's strain equivalence hypothesis [24] which was widely accepted, it is assumed that the micro-elements of rock conform to the generalized Hooke Law before rock failure. The basic relationship of rock damage constitutive equation can be established as follows:

$$\sigma = \sigma^*(1 - D) = E^*\varepsilon(1 - D), \tag{2.1}$$

where $\sigma$ refers to stress; $\sigma^*$ is the undamaged stress; $D$ denotes the damage factor; $E^*$ is the undamaged elastic modulus; and $\varepsilon$ is the strain.

Equation (2.1) can also be expressed as

$$E = E^*(1 - D), \tag{2.2}$$

where $E$ is the elastic modulus; $E^*$ is the undamaged elastic modulus; and $D$ is the damage factor.

According to equations (2.1) and (2.2), the damage factors of samples with different proportional factors (*k*) were calculated by using the parameters of peak UCS and elastic modulus, respectively, as

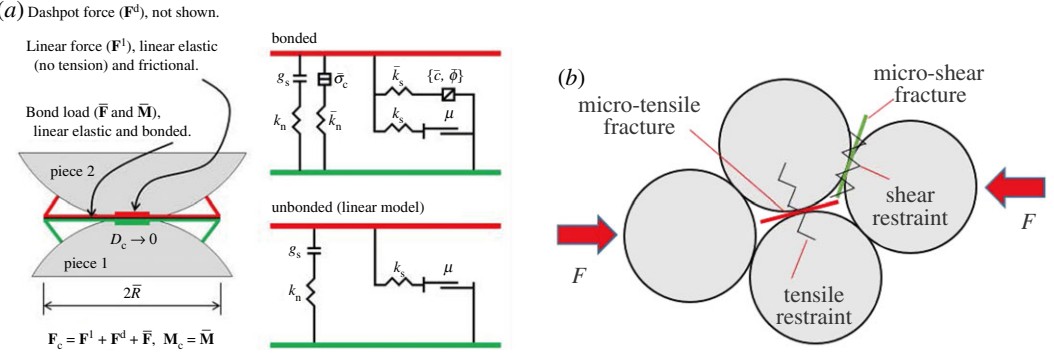

**Figure 3.** Linear parallel bond model. (*a*) Parallel bond model with inactive dashpots. (*b*) Schematic of micro fracture formation.

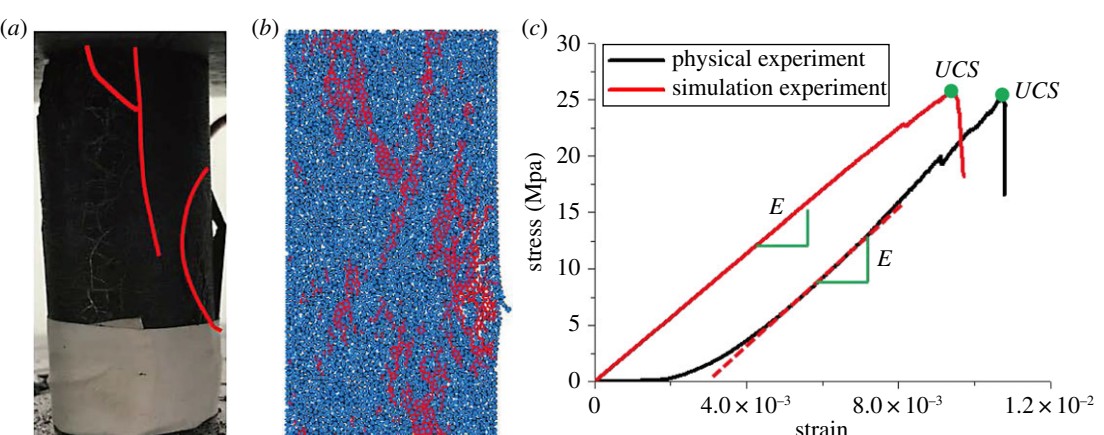

**Figure 4.** Comparison of physical experiments and simulation experiments. (*a*) Physical experiment. (*b*) Simulation experiment. (*c*) Stress−strain curve.

**Table 1.** The model parameters of linear parallel bond model.

| micro parameters | volume | micro parameters | volume |
|---|---|---|---|
| particle radius | 0.5 − 0.75 mm | effective modulus | 1.6 GPa |
| normal stiffness | 1.6 GPa | normal-to-shear stiffness ratio | 2.5 |
| normal-to-shear stiffness ratio | 2.5 | tensile strength [stress] | 8.6 MPa |
| friction angle | 45 | cohesion [stress] | 15.2 MPa |

given in table 3. The damage factor results obtained by the two parameters are very similar, and the error is controlled within 2%. Therefore, the numerical model established with different scale factors ($k$) is completely consistent with the strain equivalence hypothesis of rock damage. That is, this model can simulate rocks with different degrees of damage. Figure 6 shows the positive correlation between the damage factor $D$ and the proportional factor $k$ with a high coefficient of 99.99%. Since there is no report on the quantitative relations between the macroscopic and mesoscopic properties of the PFC model at present, it is feasible to match the peak force and the elastic modulus of rock by using the proportional factors. It can be seen from figures 7 and 8 that the energy density and tensile strength of rock failure under different damage factors are both linearly negatively correlated with damage factors.

## 3. Rock cutting model

The linear single-pick cutting model [7,26] fixes the rock and then moves the cutter to cut the rock. The rock model established in this paper was a 200 mm × 100 mm model composed of 14 420

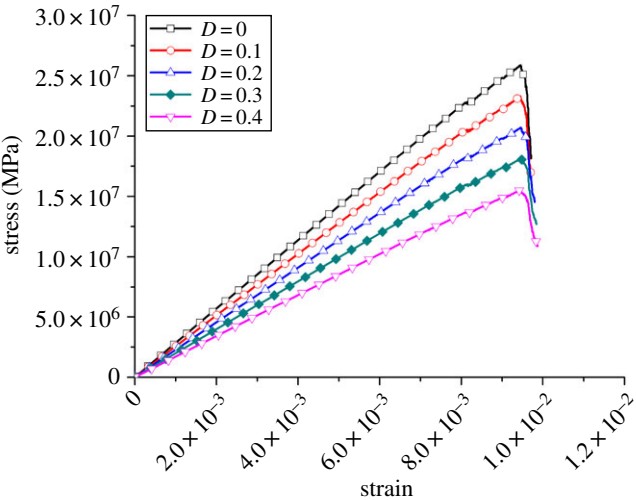

**Figure 5.** Stress–strain curves under different k.

**Table 2.** The model parameters under different k.

| micro parameters | $k = 0$ | $k = 0.1$ | $k = 0.2$ | $k = 0.3$ | $k = 0.4$ |
|---|---|---|---|---|---|
| particle radius (mm) | 0.5 – 0.75 | 0.5 – 0.75 | 0.5 – 0.75 | 0.5 – 0.75 | 0.5 – 0.75 |
| normal stiffness (GPa) | 1.6 | 1.44 | 1.28 | 1.12 | 0.96 |
| normal-to-shear stiffness ratio | 2.5 | 2.5 | 2.5 | 2.5 | 2.5 |
| friction angle | 45 | 45 | 45 | 45 | 45 |
| effective modulus (GPa) | 1.6 | 1.44 | 1.28 | 1.12 | 0.96 |
| normal-to-shear stiffness ratio | 2.5 | 2.5 | 2.5 | 2.5 | 2.5 |
| tensile strength [stress] (MPa) | 8.6 | 7.74 | 6.88 | 6.02 | 5.16 |
| cohesion [stress] (MPa) | 15.2 | 13.68 | 12.16 | 10.64 | 9.12 |

**Table 3.** Damage degree. $D_{UCS}$ is damage degree calculated by UCS; $D_M$ is damage degree calculated by elastic modulus.

| parameters | $k = 0$ | $k = 0.1$ | $k = 0.2$ | $k = 0.3$ | $k = 0.4$ |
|---|---|---|---|---|---|
| UCS/Pa | $2.59 \times 10^7$ | $2.33 \times 10^7$ | $2.07 \times 10^7$ | $1.81 \times 10^7$ | $1.55 \times 10^7$ |
| elastic modulus/Pa | $2.84 \times 10^9$ | $2.55 \times 10^9$ | $2.27 \times 10^9$ | $1.98 \times 10^9$ | $1.69 \times 10^9$ |
| $D_{UCS}$ | 0 | 0.10 | 0.20 | 0.30 | 0.40 |
| $D_M$ | 0 | 0.10 | 0.20 | 0.30 | 0.41 |
| error analysis (%) | 0.00 | 0.96 | 0.91 | 1.17 | 1.03 |
| result | $D = 0$ | $D = 0.1$ | $D = 0.2$ | $D = 0.3$ | $D = 0.4$ |

particles. The micro-parameters adopted corresponded with the sample adopted in the uniaxial compression test. That is, the rock damage factors were 0, 0.1, 0.2, 0.3 and 0.4. Meanwhile, the rock was set as a fixed displacement boundary by using the walls in PFC[2D], as shown in figure 9. The rock was cut by pickaxe cutter with an attack angle of 55°, a back rake angle of 10°, a cutting speed of 1 m s$^{-1}$ and a cutting distance of 120 mm. The cutting depths were 5, 10 and 15 mm.

In the simulation, the cutting force and the process of rock fragmentation were collected. The peak cutting force (PCF), mean cutting force (MCF) and the fluctuation of cutting force were also calculated. Among them, the fluctuation of cutting force is an important parameter of the tool load. Large fluctuations can lead to fatigue failure of the cutting tool, affecting the safety and efficiency of cutting. The parameter is mainly evaluated by the fluctuation index (FI) [27] and the pulse number (PN) [28] in unit displacement of the cutting force. FI refers to the dimensionless parameter for

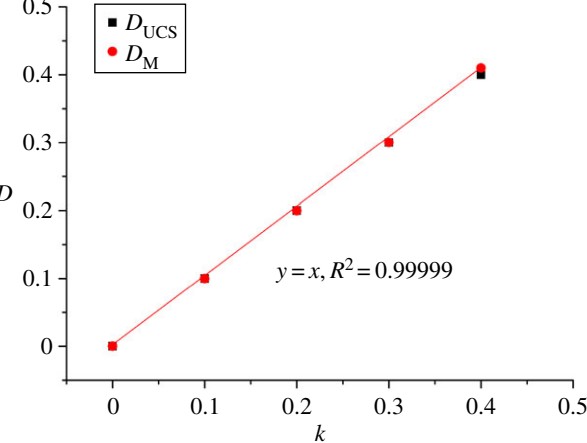

**Figure 6.** Relationships between $k$ and $D$.

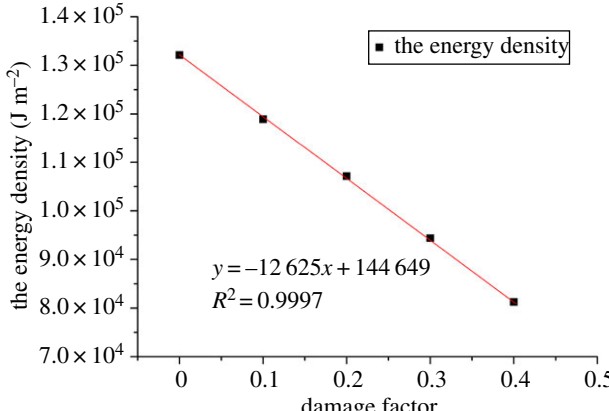

**Figure 7.** Relationships between $D$ and $E$.

evaluating the intensity of cutting force fluctuation. The larger the FI, the more violently the cutting force fluctuates. Its calculation formula is as follows:

$$\text{FI} = \frac{1}{\overline{F_C}} \cdot \frac{1}{n} \sum_{i=1}^{n} (F_{Ci} - \overline{F_C}),$$ (3.1)

where FI is the fluctuation index of cutting force; $\overline{F_C}$ is the MCF; $n$ is the number of data of cutting force; and $F_{Ci}$ is the cutting force at any time.

PN, which refers to the number of cutting force peaks within a unit cutting length, is negatively correlated with the clastic avalanche. Its calculation formula is as follows:

$$\text{PN} = \frac{m}{L},$$ (3.2)

where PN is the number of cutting force peaks within the unit cutting distance; $m$ indicates the number of peaks whose peak value is higher than a half of the mean PCF; and $L$ means the cutting distance.

In addition, the cutting consumption is analysed in the paper. Specific energy (SE) consumption is considered as a crucial index of the mining machineries. The energy consumed by cutting a unit volume of rock is described by the following equation:

$$\text{SE} = \frac{W}{V_C} = \frac{\int_0^l F_C \, dx}{V_C} = \frac{\int_0^l F_C \, dx}{S_C \times 1},$$ (3.3)

where $W$ is the integral calculation of cutting displacement through the cutting force ($F_C$); $V_C$ is the gross volume of the fragments from the rock. Because of the two-dimensional numerical model used here, the fragment area $S_C$ is used to replace $V_C$ in this paper.

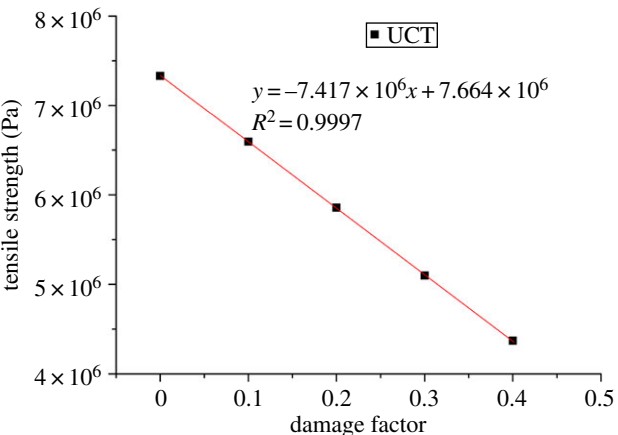

**Figure 8.** Relationships between $D$ and tensile strength.

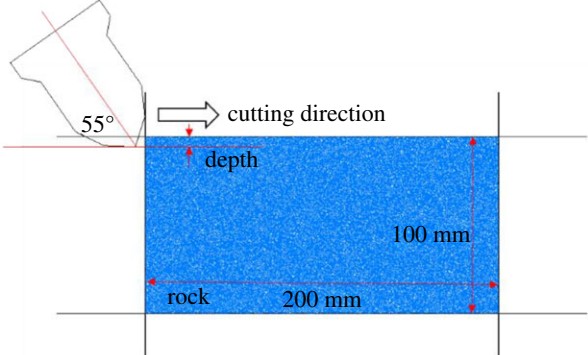

**Figure 9.** Numerical simulation model for linear single-pick cutting.

The efficiency of total energy used for cutting can be well explained by SE consumption, but $t$ friction and other forms of energy consumption exist during cutting. Aiming at precisely demonstrating the energy consumed to damage the rock in the cutting process, this paper introduces a new parameter, cutting energy utilization rate (CEUR). Given the fact that all mechanical work is used for rock failure in the uniaxial compression test, the energy density of rock failure obtained from the uniaxial compression test can be regarded as the standard quantity of rock failure energy. The CEUR is the ratio of energy density to SE

$$\text{CEUR} = \frac{\text{SE}}{E_d},$$
(3.4)

where CEUR means the utilization rate of cutting energy; and $E_d$ is the energy density of rock damage in the uniaxial compression test.

# 4. Results and discussion

## 4.1. Rock cutting failure process

The process of rock fragmentation should be grasped before a discussion of the effect of rock damage on rock cutting performance. Figure 10 exhibits the force chain network where the blue line represents the compressive stress; the green line represents the tensile stress and the thickness of the line represents the magnitude of contact stress. The bond breakage is recorded by DFN. Orange indicates the micro-tensile crack caused by tensile failure, and black indicates the shear crack caused by shear failure. Firstly, a large concentrated compressive stress is generated at the tip of the cutting pick, resulting in a crushing zone and the formation of a compact core. Then, due to the extrusion induced by the tip movement, tensile stress is generated outside the compressive stress concentration zone, while the macro crack tip propagates under the tensile stress. The initial crack roughly follows the direction of the pick axis. Since the failure mode of rock cutting is a result of tensile failure, the cutting force is greatly affected

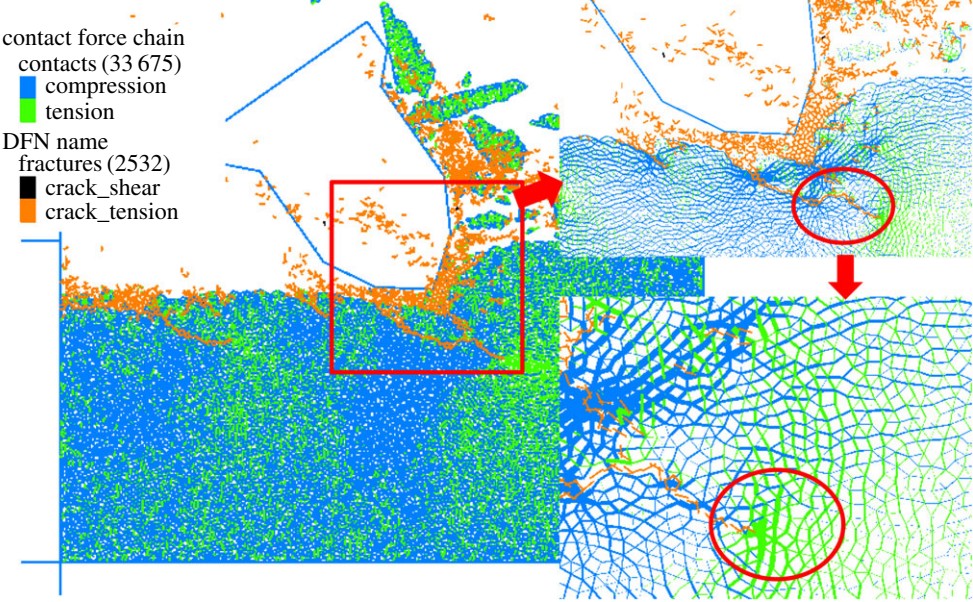

**Figure 10.** Process of rock cutting failure.

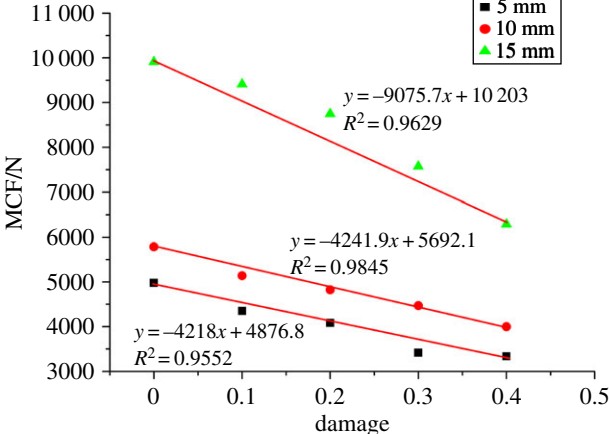

**Figure 11.** Relationship between MCF and rock damage.

by the tensile strength. The cutting force equation should be established based on the criterion of tensile stress. Without considering the crack length, $F \propto \sigma_t$ is the cutting force equation established by Li [29], that is, $F \propto -D$. Figure 11 shows the relationship between the MCF and the rock damage. An obvious linear relationship exists between the MCF and the rock damage: the MCF decreases with the increase of rock damage, which agrees with the theory.

## 4.2. Effect of damage on rock fragmentation

Figure 12 demonstrates the fragmentation patterns of rocks with different damaged rocks under different cutting depths. The red part shows the cracks caused by tensile fracture. The sample mainly undergoes tensile damage, which is consistent with the conclusion of the literature [30]. From the lateral point of view, as the cutting depth increases, not only does the blockage of rock fragmentation gradually increase, but also the number and length of cracks grow. These all demonstrate the transformation of the rock failure mode from ductile failure to brittle failure, which is in agreement with the conclusion of the literature [31]. Similarly, from the longitudinal point of view, as the degree of rock damage rises, the debris blockage of the destroyed rock gradually becomes larger, and the crack develops in advance of the cutting surface. In this case, the brittle failure becomes the predominant failure mode.

The relationship between the PCF and the damage factor is described in figure 13. Horizontally, there are three phases in the curve: The 0–0.1 level shows a relatively stable area of the PCF; the 0.1–0.3 level

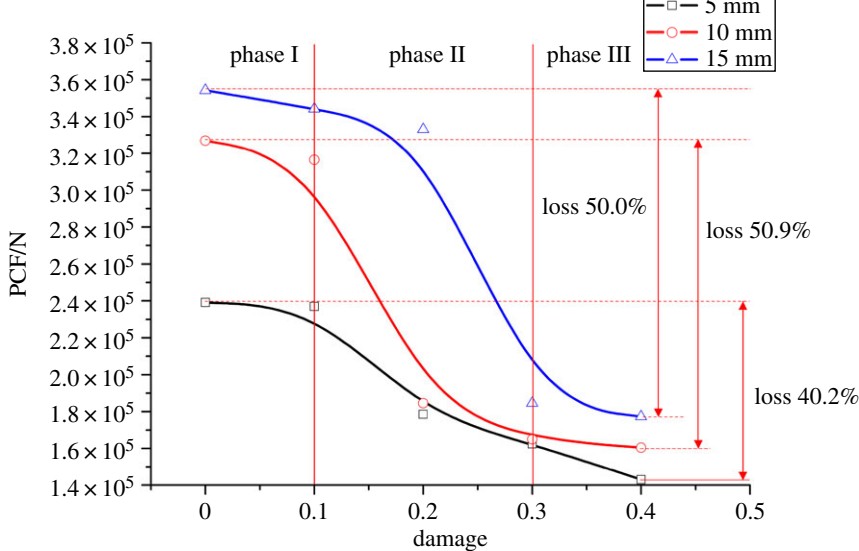

**Figure 12.** Fragmentation patterns of rocks with different damages and cutting depths.

**Figure 13.** Relationship between PCF and rock damage.

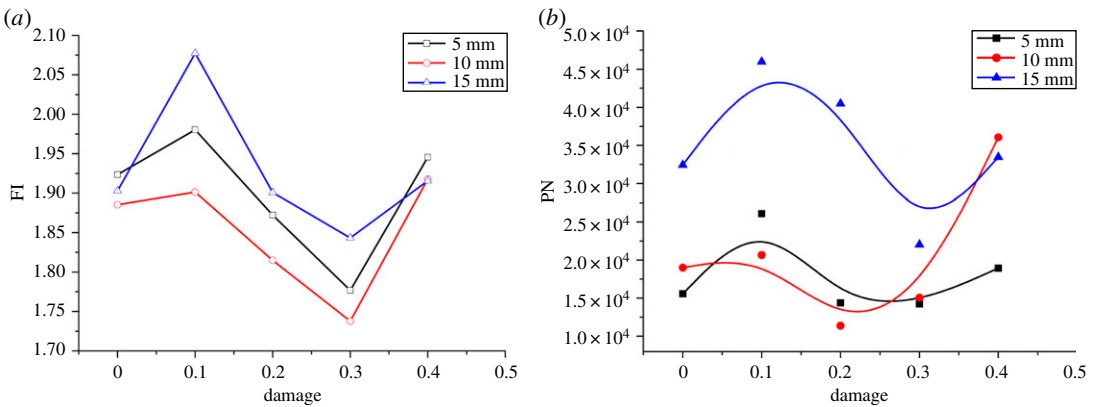

**Figure 14.** Relationship between damage and instability indexes: (*a*) FI, (*b*) PN in unit length.

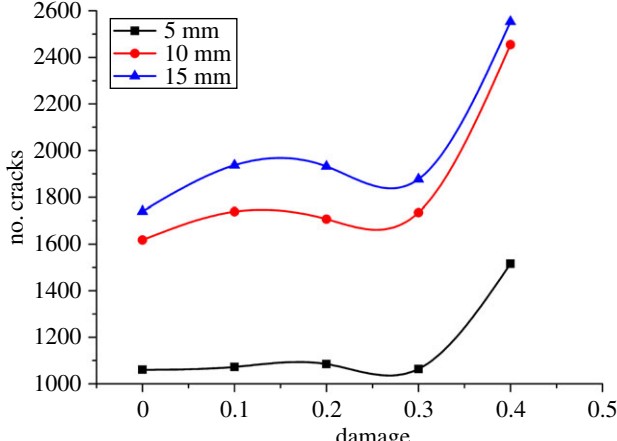

**Figure 15.** Relationship between damage and number of cracks.

demonstrates an accelerated reduction of the force; and the 0.3–0.4 level reflects a decelerated reduction of the relatively stable force. Therefore, in terms of assisted rock-breaking technology, the degree of deterioration of the rock should be at least 0.3, so as to achieve a significant reduction in the cutting force of mining machinery. Vertically, the damage factor influences the cutting force significantly. The cutting force of a sample with a damage factor of 0.4 is 177 kN and 50% lower than that of the intact sample. Figure 14 shows the relationship between the cutting force fluctuation and the damage factor. Both FI and PN increase first, then drop, and finally rise again with the increase of the damage factor. When the damage factor is small, large deformation is allowed to occur in rock cutting. As a result, the crack increases gradually during deformation, resulting in a stronger cutting force fluctuation. However, when the rock is damaged to a certain extent, large deformation is not allowed to occur in rock cutting. As the tool advances, the fracturing occurs continuously, which enhances the stability and reduces the fluctuation of the cutting force. When the rock damage continues to increase, the strength of the rock decreases, and the existence of unbalanced force induces the generation of more cracks in the rock, which results in a sudden increase of cutting cracks and increases the instability of cutting. The above analysis is reflected in figures 13 and 15. From the perspective of cutting stability, a rock damage factor of 0.3 facilitates the cutting performance most.

## 4.3. Effect of rock damage on specific energy

Figure 16 exhibits the negative relationship between the cutting SE consumption and the damage factor. It can be observed from figure 16 that the cutting energy consumption decreases linearly with the increase of rock damage factor. The reason is as follows: when the damage factor is small, tensile failure is the main failure mode, and the formation of fine debris consumes a lot of energy. With the increase of rock damage factor, rock fragmentation transforms to brittle failure, and macro cracks

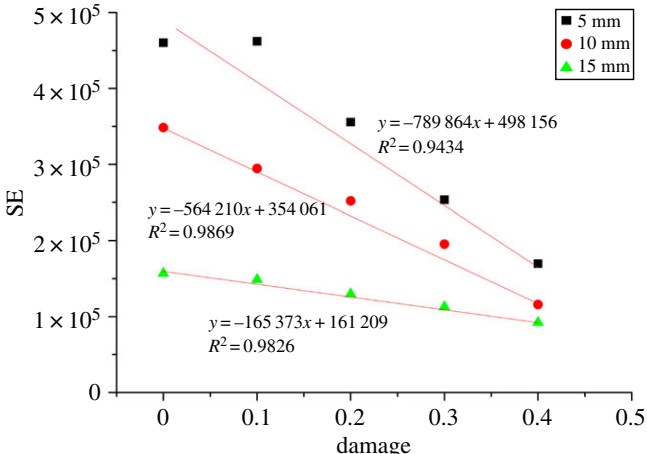

**Figure 16.** Relationship between damage and SE.

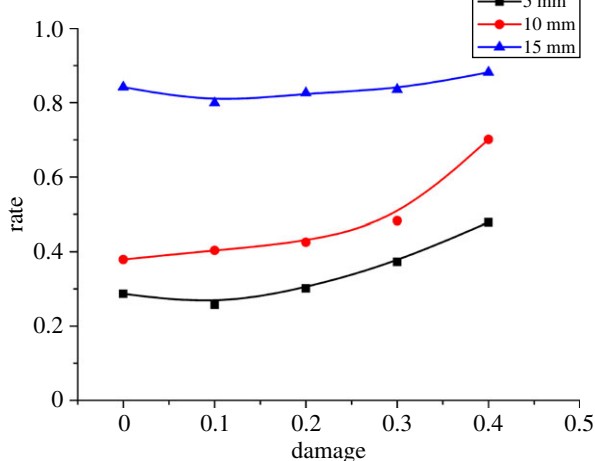

**Figure 17.** Relationship between damage and CEUR.

expand and form larger debris. This damage reduces SE consumption. Figure 17 shows the relationship between the CEUR and the damage factor. The CEUR, which is the utilization rate of cutting energy, rises with the increase of damage factor. Nevertheless, the CEUR do not show significant changes when the cutting depth becomes large, because friction energy consumption is the main factor affecting CEUR. With the increase of damage factor, cracks generated by the rock cutting develop and expand prematurely, along with the failure of large particles, which reduces the friction energy consumption and improves the cutting energy efficiency. When the cutting depth is large, the continuous crack propagation and bulk debris peeling (figure 8) weakens the impact of damage factor on cutting energy efficiency.

## 5. Conclusion

In this paper, the discrete element software PFC$^{2D}$ was used to study the influence of rock damage degree on rock fracturing and cutting performance during rock cutting. The main conclusions are as follows.

Firstly, the uniaxial compression test proves that it is logical and reasonable to simulate rock with different damage factors by reducing the elastic modulus, the tensile strength and the shear strength according to the proportional factor. Besides this, there is a significant and reliable linear relationship between the proportional factor and the damage factor, with the correlation coefficient being 0.9999. Since there is no report on the quantitative relations between macroscopic and mesoscopic properties of the PFC model at present, it is feasible to match the peak force and the elastic modulus of rock by using the proportional factors.

Secondly, in terms of rock fragmentation, the dominant role of tensile mode does not change with different damage factors in rock failure. However, as the damage factor increases, the rock damage mode gradually transforms from tensile failure to brittle failure. Meanwhile, the debris blockage of the destroyed rock becomes significantly larger, and the microcracks increase in number and develop in advance of the cutting surface.

Thirdly, the analysis of cutting force suggests that the degree of rock damage is one of the key factors affecting the cutting force. The PCF decreases obviously as the rock damage factor grows within the range of 0.1–0.3. In addition, a clearly negative linear correlation exists between the average cutting force and the damage factor. Therefore, this study is of great significance for the development of assisted rock-breaking technologies. The technology can weaken the rock cutting force. However, when the damage factor of rock is small, the effect of assisted rock breaking is not obvious. It is suggested that the damage of rock by assisted rock-breaking technologies should reach the damage factor of 0.3. Accordingly, the analyses of FI and PN demonstrate that the cutting force does not fluctuate much when the damage factor is 0.3, which facilitates the reliability of cutting tools.

Finally, as one of the important ways to improve the cutting efficiency, the rock damage degree can not only reduce the SE consumption of rock cutting, but also raise the utilization rate of cutting energy consumption. The conclusions of this paper are of great significance for the design of cutting head layout of excavation machinery based on the rock damage degree. Taking the design of shearer cutting drums as an example, the design of different damage degrees of coal in the footage range can be taken into account in the design of cutting pick pitch, so as to improve cutting performance and lump coal rate and increase production capacity.

Data accessibility. Data available from the Dryad Digital Repository: https://doi.org/10.5061/dryad.46pv990 [32].
Authors' contributions. S.Liu, S.Lu and Z.W. conceived the study design; S.Liu and S.Lu performed the numerical simulation; S.Liu analysed the data and wrote the manuscript; Z.W. and J.C. helped to edit the manuscript.
Competing interests. We declare we have no competing interests.
Funding. This research was supported by 'The Fundamental Research Funds for the Central Universities' (2017CXNL01).
Acknowledgments. We thank Jingchao Wang for their support of our study.

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
