## [Reviewer comments · Royal Society Open Science]

Review History

RSOS-190116.R0 (Original submission)

Review form: Reviewer 1

Is the manuscript scientifically sound in its present form?

No

Are the interpretations and conclusions justified by the results?

No

Is the language acceptable?

Yes

Is it clear how to access all supporting data?

No

Do you have any ethical concerns with this paper?

No

Have you any concerns about statistical analyses in this paper?

No

Recommendation?

Reject

Comments to the Author(s)

1. I cannot find in this manuscript how to define the factor k , and how to relate the k and damage factor D . Please give an explanation.
2. The microparameters in PFC model are very important for the calculation to reveal the real mechanism of the coaling process. Though the authors has used the uniaxial tests to determine the parameters, however, I cannot find the proof such as the testing curves or test conditions.
3. 4.1 and 4.2 is the same title.
4. How to modeling the cutting depth in PFC 2D? Is the depth is firstly supposed as a value, and then a drilling rate is given on the rock? It is not real for the drilling process. And from the Fig 2 and Fig.7, the drilling processes are different.
5. Check the figure 4, the curves are linear, it is not correct. And also, in Fig.5 and Fig.6, the relationship of the k And D , D and E , why are the two parameters are equal?
6. The rock breakage is very complex because of rock properties, drilling process and corresponding factors, we should find the breakage process based on the clear mechanical parameters to reveal the breakage mechanism. The description in this manuscript gives no details.

Review form: Reviewer 2

Is the manuscript scientifically sound in its present form?

No

Are the interpretations and conclusions justified by the results?

Yes

Is the language acceptable?

No

Is it clear how to access all supporting data?

Not Applicable

Do you have any ethical concerns with this paper?

No

Have you any concerns about statistical analyses in this paper?

No

Recommendation?

Major revision is needed (please make suggestions in comments)

Comments to the Author(s)

The authors investigate the effect of prior inherited damage by the rock on its resistance to mechanical cutting with cutting picks usually met on the cutting heads of roadheaders. For this

investigation they rely on primitive concepts of damage mechanics and commercial numerical software (PFC2d) to simulate the action of a single pick during rock cutting. Significant improvement of the use of the English language is required before the possible publication of the paper. Further comments are listed below:

- In p. 1. The sentence "Two major working procedures are needed in mining: to excavate the mining channel and the ore body mining, as shown in Fig. 1. .." is not correct both in mining terms and with regards to its grammar or syntax .
- In the Introduction and in the literature review the authors should mention the relevant paper regarding simulation of rock cutting with PFC and other methods by M. Stavropoulou, Modeling of small-diameter rotary drilling tests on marbles, *International Journal of Rock Mechanics & Mining Sciences* 43 (2006) 1034–1051, also with regards to rock damage the paper: Exadaktylos G. and Stavropoulou M., (2008) A Specific Upscaling Theory of Rock Mass Parameters Exhibiting Spatial Variability: Analytical relations and computational scheme, *International Journal of Rock Mechanics and Mining Sciences*, 45 (2008) pp. 1102–1125.

- The damage influences the 7 parameters of the particle model of the rock that are involved in the simulation, namely the two stiffnesses, UCS, UTS, friction angle.
- In p. 3 the authors describe the cutting process but they do not mention the attack angle and the back rake angle..."7. The rock was cut by pickaxe cutter with an angle of 55 degrees, a cutting speed of 1 m/s and a cutting distance of 120 mm. The cutting depth is 5mm, 10mm and 15mm....".
- Since the author have found from the simulations that the rock breakage is mainly performed through the creation of tensile cracks why did they have chose the Uniaxial Compression test (UCT) for the calibration of damage and for forming the ratio of SE over the energy expended in this test up to the point of failure instead of the uniaxial tensile test?

The paper is concerned with a topic of appreciable practical significance, namely that of pre-conditioning the rock for subsequent action of a mechanical cutting pick that will result into more efficient cutting processes. However, the paper needs significant improvement before its publication.

Review form: Reviewer 3

Is the manuscript scientifically sound in its present form?

Yes

Are the interpretations and conclusions justified by the results?

Yes

Is the language acceptable?

Yes

Is it clear how to access all supporting data?

Yes

Do you have any ethical concerns with this paper?

No

Have you any concerns about statistical analyses in this paper?

No

Recommendation?

Major revision is needed (please make suggestions in comments)

Comments to the Author(s)

This paper investigated the rock fragmentation induced by rock cutting via DEM with PBM.

Some comments are the following:

- (1) The definition of parameter k was not clear, though it is a very crucial parameter in this paper.
- (2) To my knowledge, the damage factor reflect the damage degree, i.e., the more the micro crack or bond damage is, the larger the damage factor is. Therefore, the DEM samples with different damage factors should be prepared according to their micro characteristics. However in this paper, only the different of E and UCS are provided for samples with different damage factors, whereas the description of the micro characteristic was not found.
- (3) The macro parameters, e.g., the compressive and tensile strength, should be provided.
- (4) The titles of sub-sections 4.1 and 4.2 were the same in the manuscript.
- (5) The investigation of influence of rock cutting on fragmentation should be more detailed, and more micro information, e.g. the force chain, the bond breakage, is expected to be provided. The content in section 4 was too little.
- (6) The English needs improvement.
- (7) “、” should be revised as “,” at Line 48, first page.

Major revision is needed.

Decision letter (RSOS-190116.R0)

12-Feb-2019

Dear Dr Liu,

The editors assigned to your paper ("Investigation for the influence mechanism of rock damage on rock fragmentation and cutting performance by discrete element method") have now received comments from reviewers. We would like you to revise your paper in accordance with the referee and Associate Editor suggestions which can be found below (not including confidential reports to the Editor). Please note this decision does not guarantee eventual acceptance.

Please submit a copy of your revised paper before 07-Mar-2019. Please note that the revision deadline will expire at 00.00am on this date. If we do not hear from you within this time then it will be assumed that the paper has been withdrawn. In exceptional circumstances, extensions may be possible if agreed with the Editorial Office in advance. We do not allow multiple rounds of revision so we urge you to make every effort to fully address all of the comments at this stage. If deemed necessary by the Editors, your manuscript will be sent back to one or more of the original reviewers for assessment. If the original reviewers are not available, we may invite new reviewers.

- Data accessibility

If you wish to submit your supporting data or code to Dryad (<http://datadryad.org/>), or modify your current submission to dryad, please use the following link:
<http://datadryad.org/submit?journalID=RSOS&manu=RSOS-190116>

- Competing interests

- Authors' contributions

- Acknowledgements

- Funding statement

on behalf of Prof R. Kerry Rowe (Subject Editor)
openscience@royalsociety.org

Associate Editor's comments:

Three reviewers have reported on your manuscript. Each has identified substantial areas of weakness in your paper, but have provided recommendations of how to improve the paper and make it fit for publication. Please ensure you incorporate their recommendations, including seeking the support of a language polishing service (<https://royalsociety.org/journals/authors/language-polishing/>). If you do not incorporate their recommendations, you must provide a fully reasoned rebuttal for this. Furthermore, you must include evidence of having had your manuscript reviewed by a language polishing service. Your manuscript will likely return to these reviewers for further assessment: if you don't persuade them that the paper is ready for publication, it will be rejected from further consideration.

Comments to Author:

Reviewers' Comments to Author:
Reviewer: 1

Comments to the Author(s)

1. I cannot find in this manuscript how to define the factor k , and how to relate the k and damage factor D . Please give an explanation.
2. The microparameters in PFC model are very important for the calculation to reveal the real mechanism of the coaling process. Though the authors has used the uniaxial tests to determine the parameters, however, I cannot find the proof such as the testing curves or test conditions.
3. 4.1 and 4.2 is the same title.
4. How to modeling the cutting depth in PFC 2D? Is the depth is firstly supposed as a value, and then a drilling rate is given on the rock? It is not real for the drilling process. And from the Fig 2 and Fig.7, the drilling processes are different.
5. Check the figure 4, the curves are linear, it is not correct. And also, in Fig.5 and Fig.6, the relationship of the k And D , D and E , why are the two parameters are equal?
6. The rock breakage is very complex because of rock properties, drilling process and corresponding factors, we should find the breakage process based on the clear mechanical parameters to reveal the breakage mechanism. The description in this manuscript gives no details.

Reviewer: 2

Comments to the Author(s)

The authors investigate the effect of prior inherited damage by the rock on its resistance to mechanical cutting with cutting picks usually met on the cutting heads of roadheaders. For this investigation they rely on primitive concepts of damage mechanics and commercial numerical software (PFC2d) to simulate the action of a single pick during rock cutting. Significant improvement of the use of the English language is required before the possible publication of the paper. Further comments are listed below:

- In p. 1. The sentence "Two major working procedures are needed in mining: to excavate the mining channel and the ore body mining, as shown in Fig. 1. ..." is not correct both in mining terms and with regards to its grammar or syntax .
- In the Introduction and in the literature review the authors should mention the relevant paper regarding simulation of rock cutting with PFC and other methods by M. Stavropoulou, Modeling of small-diameter rotary drilling tests on marbles, International Journal of Rock Mechanics & Mining Sciences 43 (2006) 1034–1051, also with regards to rock damage the paper: Exadaktylos G. and Stavropoulou M., (2008) A Specific Upscaling Theory of Rock Mass Parameters Exhibiting Spatial Variability: Analytical relations and computational scheme, International Journal of Rock Mechanics and Mining Sciences, 45 (2008) pp. 1102–1125.

- The damage influences the 7 parameters of the particle model of the rock that are involved in the simulation, namely the two stiffnesses, UCS, UTS, friction angle.
- In p. 3 the authors describe the cutting process but they do not mention the attack angle and the back rake angle..."7. The rock was cut by pickaxe cutter with an angle of 55 degrees, a cutting speed of 1 m/s and a cutting distance of 120 mm. The cutting depth is 5mm, 10mm and 15mm....".
- Since the authors have found from the simulations that the rock breakage is mainly performed through the creation of tensile cracks why did they have chosen the Uniaxial Compression test (UCT) for the calibration of damage and for forming the ratio of SE over the energy expended in this test up to the point of failure instead of the uniaxial tensile test?

The paper is concerned with a topic of appreciable practical significance, namely that of pre-conditioning the rock for subsequent action of a mechanical cutting pick that will result into more efficient cutting processes. However, the paper needs significant improvement before its publication.

Reviewer: 3

Comments to the Author(s)

This paper investigated the rock fragmentation induced by rock cutting via DEM with PBM.

Some comments are the following:

- (1) The definition of parameter k was not clear, though it is a very crucial parameter in this paper.
- (2) To my knowledge, the damage factor reflects the damage degree, i.e., the more the micro crack or bond damage is, the larger the damage factor is. Therefore, the DEM samples with different damage factors should be prepared according to their micro characteristics. However in this paper, only the difference of E and UCS are provided for samples with different damage factors, whereas the description of the micro characteristic was not found.
- (3) The macro parameters, e.g., the compressive and tensile strength, should be provided.
- (4) The titles of sub-sections 4.1 and 4.2 were the same in the manuscript.
- (5) The investigation of influence of rock cutting on fragmentation should be more detailed, and more micro information, e.g. the force chain, the bond breakage, is expected to be provided. The content in section 4 was too little.
- (6) The English needs improvement.
- (7) “、” should be revised as “,” at Line 48, first page.

Major revision is needed.

Author's Response to Decision Letter for (RSOS-190116.R0)

See Appendix A.

RSOS-190116.R1 (Revision)

Review form: Reviewer 1

Is the manuscript scientifically sound in its present form?

Yes

Are the interpretations and conclusions justified by the results?

Yes

Is the language acceptable?

Yes

Is it clear how to access all supporting data?

Yes

Do you have any ethical concerns with this paper?

No

Have you any concerns about statistical analyses in this paper?

No

Recommendation?

Accept as is

Comments to the Author(s)

The corrections are acceptable.

Review form: Reviewer 3

Is the manuscript scientifically sound in its present form?

Yes

Are the interpretations and conclusions justified by the results?

Yes

Is the language acceptable?

Yes

Is it clear how to access all supporting data?

Yes

Do you have any ethical concerns with this paper?

No

Have you any concerns about statistical analyses in this paper?

No

Recommendation?

Accept as is

Comments to the Author(s)

The authors revised the manuscript according to the comments. All the comments were responded, and the revision was well done. I therefore recommend this paper to be published.

Decision letter (RSOS-190116.R1)

01-Apr-2019

Dear Dr Liu,

I am pleased to inform you that your manuscript entitled "Investigation for the influence mechanism of rock damage on rock fragmentation and cutting performance by the discrete element method" is now accepted for publication in Royal Society Open Science.

on behalf of Professor R. Kerry Rowe (Subject Editor)
openscience@royalsociety.org

Reviewer comments to Author:

Reviewer: 1

Comments to the Author(s)

The corrections are acceptable.

Reviewer: 3

Comments to the Author(s)

The authors revised the manuscript according to the comments. All the comments were responded, and the revision was well done. I therefore recommend this paper to be published.

Appendix A

Revision response to Associate Editor

Response to the comments of associate editor of the paper entitled “ Investigation for the influence mechanism of rock damage on rock fragmentation and cutting performance by the discrete element method”

Many thanks for constructive comments from the reviewers and the editor. These comments and suggestions have been carefully considered in the current manuscript. Those revised texts were labelled in red in the manuscript. Major changes are summarized as below:

Remark #1: you must include evidence of having had your manuscript reviewed by a language polishing service.

Action: Done.

Revision response to Reviewer #1

Response to the comments of Reviewer 1 of the paper entitled “ Investigation for the influence mechanism of rock damage on rock fragmentation and cutting performance by the discrete element method”

Many thanks for constructive comments from the reviewers and the editor. These comments and suggestions have been carefully considered in the current manuscript. Those revised texts were labelled in red in the manuscript. Major changes are summarized as below:

Remark #1: I cannot find in this manuscript how to define the factor k , and how to relate the k and damage factor D . Please give an explanation.

Action: Done. Thanks for comments. Factor k , a newly introduced variable, is a scale factor used to adjust the microscopic parameters in the PFC model and can be called a microscopic variable. D , the damage variable in the mechanical parameters, is a macroscopic variable. Before the uniaxial compression experiment, there is no quantitative relationship between k and D . It was found through experimental determination and calculation that k and D were equal. Therefore, k and D are variables in different scales, but their values are equal.

Remark #2: The micro-parameters in PFC model are very important for the calculation to reveal the real mechanism of the coaling process. Though the authors has used the uniaxial tests to determine the parameters, however, I cannot find the proof such as the testing curves or test conditions.

Action: Done. The results of uniaxial compression test are added, as shown in Figure 4, and details of the PFC model calibration are also shown in the figure.

Remark #3: 4.1 and 4.2 is the same title.

Action: Done. We are sorry for the misunderstanding caused by writing.

Remark #4: How to modeling the cutting depth in PFC 2D? Is the depth is firstly supposed as a value, and then a drilling rate is given on the rock? It is not real for the

drilling process. And from the Fig 2 and Fig.7, the drilling processes are different.

Action: Done. In this paper, the depth is pre-determined and the pick is placed in the corresponding position. Then the pick is given a speed to cut the rock.

Regarding the relationship between Figures 2 and 7, we studied a single pick from multiple picks. Generally, the cutting pathway of a single pick can be simplified as a straight line, because a cutting unit of the arc pathway could be deemed as a line in experimental scale modeling.

Remark #5: Check the figure 4, the curves are linear, it is not correct. And also, in Fig.5 and Fig.6, the relationship of the k And D, D and E, why are the two parameters are equal?

Action: Done. Generally, PFC simulation starts from the elastic stage of rock, because PFC model cannot simulate the non-linear deformation of rock caused by the compression and closure of internal cracks.

The factor k is a microscopic parameter and D is a macroscopic parameter. There is neither any quantitative relationship between microscopic parameters and macroscopic parameters in PFC nor any relevant report. The equivalence of k and D is acquired from the experimental results.

Figure 6 shows the relationship between the energy density and the rock damage factor D.

Remark #6: The rock breakage is very complex because of rock properties, drilling process and corresponding factors, we should find the breakage process based on the clear mechanical parameters to reveal the breakage mechanism. The description in this manuscript gives no details.

Action: Done. According to your suggestion, this paper presents the process of rock fragmentation by analyzing the force chain network and crack propagation near the pick. In addition, the influence of damage factor on cutting force is analyzed. Please check Section 4.1.

Revision response to Reviewer #2

Response to the comments of reviewer 2 of the paper titled “ Investigation for the influence mechanism of rock damage on rock fragmentation and cutting performance by the discrete element method”

Many thanks for constructive comments from the reviewers and the editor. These comments and suggestions have been carefully considered in the current manuscript. The revised texts were marked in red in the manuscript. Major changes are summarized as below:

Remark #1: In p. 1. The sentence "Two major working procedures are needed in mining: to excavate the mining channel and the ore body mining, as shown in Fig. 1. ..." is not correct both in mining terms and with regards to its grammar or syntax .

Action: Done. Errors have been corrected after polishing the language.

Remark #2: In the Introduction and in the literature review the authors should mention the relevant paper regarding simulation of rock cutting with PFC and other methods by M. Stavropoulou, Modeling of small-diameter rotary drilling tests on marbles, International Journal of Rock Mechanics & Mining Sciences 43 (2006) 1034–1051, also with regards to rock damage the paper: Exadaktylos G. and Stavropoulou M., (2008) A Specific Upscaling Theory of Rock Mass Parameters Exhibiting Spatial Variability: Analytical relations and computational scheme, International Journal of Rock Mechanics and Mining Sciences, 45 (2008) pp. 1102–1125.

Action: Done. We did not notice the paper “Modeling of Small-diameter Rotary Drilling Tests on Marbles” at first because it is a paper on drilling instead of cutting. However, after the reviewer’s reminder, we find that the content of PFC in this paper is very valuable for reference, and it is quoted.

“A Specific Upscaling Theory of Rock Mass Parameters Exhibiting Spatial Variability: Analytical Relations and Computational Scheme” is a paper of high quality which has a reference value for the method of damage factor calculation in the manuscript, so it is cited.

Remark #3: The damage influences the 7 parameters of the particle model of the rock that are involved in the simulation, namely the two stiffnesses, UCS, UTS, friction angle.

Action: Done. There are 7 parameters in PFC5.0. They are the elastic modulus of linear contact, the stiffness ratio of linear contact, the elastic modulus of parallel bond, the stiffness ratio of parallel bond, the tensile strength of parallel bond, the shear strength of parallel bond and the friction angle. However, the previous version only has 5 parameters, namely the two stiffnesses, the UCS, the UTS and the friction angle.

Remark #4: In p. 3 the authors describe the cutting process but they do not mention the attack angle and the back rake angle..."7. The rock was cut by pickaxe cutter with an angle of 55 degrees, a cutting speed of 1 m/s and a cutting distance of 120 mm. The cutting depth is 5mm, 10mm and 15mm...".

Action: The attack angle and the back rake angle have been added. The rock was cut by pickaxe cutter with an attack angle of 55 degrees, a back rake angle of 10 degrees, a cutting speed of 1 m/s and a cutting distance of 120 mm.

Remark #5: Since the author have found from the simulations that the rock breakage is mainly performed through the creation of tensile cracks why did they have chose the Uniaxial Compression test (UCT) for the calibration of damage and for forming the ratio of SE over the energy expended in this test up to the point of failure instead of the uniaxial tensile test?

Action: Done. As shown in Figure 10 in the manuscript, it is true that tensile failure occurs in micro-scale during rock cutting. This tensile stress is produced by the squeezing action and is not a direct tensile action. The tensile failure is similar to the compression-induced tensile failure in the uniaxial compression test. This is the reason why the uniaxial compression experiment is used.

Remark #6: The paper is concerned with a topic of appreciable practical significance, namely that of pre-conditioning the rock for subsequent action of a mechanical cutting

pick that will result into more efficient cutting processes. However, the paper needs significant improvement before its publication.

Action: Done.

Revision response to Reviewer #3

Response to the comments of reviewer 3 of the paper titled “ Investigation for the influence mechanism of rock damage on rock fragmentation and cutting performance by the discrete element method”

Many thanks for the constructive comments from the reviewers and the editor. These comments and suggestions have been carefully considered in the current manuscript. Those revised texts were marked in red in the manuscript. Major changes are summarized as below:

Remark #1: The definition of parameter k was not clear, though it is a very crucial parameter in this paper.

Action: Done. Thanks for comments. Factor k , a new variable introduced, is a scaling factor used to adjust the microscopic parameters in the PFC model, and it has no practical physical significance.

Remark #2: To my knowledge, the damage factor reflect the damage degree, i.e., the more the micro crack or bond damage is, the larger the damage factor is. Therefore, the DEM samples with different damage factors should be prepared according to their micro characteristics. However in this paper, only the different of E and UCS are provided for samples with different damage factors, whereas the description of the micro characteristic was not found.

Action: Done. The microcosmic parameters reduced by the scale factor are shown in Table 2.

Remark #3: The macro parameters, e.g., the compressive and tensile strength, should be provided.

Action: Done. The uniaxial compressive strength is shown in Table 3 and the tensile strength is shown in Fig. 8.

Remark #4: The titles of sub-sections 4.1 and 4.2 were the same in the manuscript.

Action: Done. We are sorry for the misunderstanding caused by editing.

Remark #5: The investigation of influence of rock cutting on fragmentation should be more detailed, and more micro information, e.g. the force chain, the bond breakage, is expected to be provided. The content in section 4 was too little.

Action: Done. We have analyzed the process of rock fragmentation by using the force chain and the bond breakage in Section 4.1.

Remark #6: The English needs improvement.

Action: Done. The language of the manuscript has been polished.

Remark #7: “、” should be revised as “,” at Line 48, first page.

Action: Done.